# Emergence of *Cfr*-Mediated Linezolid Resistance among Livestock-Associated Methicillin-Resistant *Staphylococcus aureus* (LA-MRSA) from Healthy Pigs in Portugal

**DOI:** 10.3390/antibiotics11101439

**Published:** 2022-10-19

**Authors:** Célia Leão, Lurdes Clemente, Maria Cara d’Anjo, Teresa Albuquerque, Ana Amaro

**Affiliations:** 1Laboratory of Bacteriology and Mycology, National Institute of Agrarian and Veterinary Research (INIAV, IP), 2780-157 Oeiras, Portugal; 2MED-Mediterranean Institute for Agriculture, Environment and Development, 7006-554 Évora, Portugal; 3CIISA-Centre for Interdisciplinary Research in Animal Health, Faculty of Veterinary Science, 1300-477 Lisbon, Portugal; 4DGAV-Portuguese National Authority for Animal Health, 1700-093 Lisbon, Portugal

**Keywords:** LA-MRSA, pigs, linezolid resistance, *cfr* gene, WGS, ST398

## Abstract

Livestock-associated methicillin-resistant *Staphylococcus aureus* (LA-MRSA) ST398 is mainly found in Europe and North America, colonizing the nasal cavity of pigs. This study characterized the MRSA isolates recovered from pig nasal swabs (*n* = 171) by evaluating the antimicrobial susceptibility profile by broth microdilution and characterizing the genetic lineages by *spa*-typing. Three linezolid-resistant isolates were subjected to Whole-Genome Sequencing (WGS). All strains harbored the *mecA* gene and were resistant to tetracycline and susceptible to vancomycin. A high frequency of multidrug resistance (97.6%) was evidenced, with 55 different multidrug resistance profiles identified. The MRSA strains were found to belong to 17 *spa*-types, three being novel. The linezolid-resistant strains appeared to belong to the ST398 type, *spa*-type t011, and SCC*mec*_type_Vc and to harbor the *cfr*, *fex*A, *bla*Z, *mec*A, *tet*M, and *tet*K genes. The *cfr* gene was predicted to be carried in the plasmid, flanked by *IS*Sau9 and the transposon TnpR. MRSA from Portuguese fattening pigs present a high diversity of genetic lineages. The presence of *cfr*-positive LA-MRSA may represent a risk of transmission to humans, mainly to those in contact with livestock.

## 1. Introduction

*Staphylococcus aureus* is a widely disseminated commensal organism colonizing the nasal mucosa and skin of humans and livestock such as pigs, cattle, poultry, pets, and wild animals [1,2,3,4,5,6]. Therefore, *S. aureus* can also be an opportunistic pathogen responsible for many human infections, from mild skin infections to life-threatening diseases [7]. In animals, it can also cause a variety of infections such as mastitis and udder impetigo, pyoderma in adults and neonates, omphalitis, arthritis, and several other pyemic conditions [8].

Methicillin-resistant *S. aureus* (MRSA) has been colonizing and infecting humans and animals [1,7,8]. Among MRSA associated with livestock (LA-MRSA) and able to colonize humans, the MRSA ST398 grouping within the clonal complex (CC) 398 is reported globally and is widespread in Europe and North America [9,10]. LA-MRSA CC398 poses a zoonotic risk, particularly for those working in close contact with livestock, such as farmers, veterinarians, abattoir workers, and people living in areas with a high livestock density [9,10,11].

Several studies reported that LA-MRSA CC398 evolved from the human methicillin-susceptible *S. aureus* (MSSA) CC398 by losing some virulence features (such as ØSa3 prophage and Panton–Valentine leukocidin, PVL) and acquiring new virulence and resistance genes (copper and zinc *czr*C, tetracycline *tet*(M) and β-lactam *mec*A and *bla*Z) [12]. LA-MRSA CC398 also acquired other antimicrobial resistance genes from Gram-positive and Gram-negative bacteria by plasmid-mediated horizontal gene transfer, including the gene for the phenicol exporter *fex*A, tetracycline *tet*L, macrolide, lincosamide, and streptogramin B *erm*T, trimethoprim *dfr*K, ABC transporters *vga*C and *vga*E, and chloramphenicol-florfenicol *cfr* [12]. The *cfr* gene encodes an rRNA methyltransferase targeting an adenine residue in the 23S rRNA (A2503), which apart from phenicols, also confers resistance to clindamycin, pleuromutilins, streptogramin A, and oxazolidinones [12,13]. This latter antibiotic class includes linezolid, approved for treating hospital-acquired pneumonia caused by *S. aureus*, including the MSSA and MRSA strains, multidrug-resistant *Streptococcus pneumoniae,* and vancomycin-resistant *Enterococcus faecium* (VRE) infections, among others [13].

In Portugal, MRSA strains belonging to CC398 have been identified in swine farms, wild boars, pets, wild rabbits, quails, and quail meat [3,4,5,6]. In addition, the first detection of linezolid resistance encoded by the *cfr* gene occurred in 2019 in three MRSA isolates recovered from infected human wounds, but none were included in the CC398 lineage [14]. In 2019, under the scope of MRSA monitoring from healthy pigs at slaughter in Portugal, three linezolid-resistant MRSA isolates belonging to the CC398 cluster and encoded by the acquired resistance *cfr* gene were detected [15]. Nevertheless, linezolid-resistant MRSA was firstly reported in one dog with severe bilateral otitis, although the genetic basis of resistance was not determined [16].

The emergence of linezolid-resistant MRSA CC398 in food-producing animals carrying the *cfr* gene poses a significant public health concern due to the possible spread of these bacteria through the animal–human interface by sharing genes through mobile genetic elements. Thus, this study aimed to characterize further the MRSA population previously reported [15], focusing on typing and antibiotic resistance profiles. Moreover, the three linezolid-resistant LA-MRSA isolates carrying the *cfr* gene were analyzed by Whole-Genome Sequencing (WGS).

## 2. Results

### 2.1. Bacterial Isolates and Antimicrobial Resistance Patterns

From 171 samples, 169 isolates (98.8%) were identified as MRSA carrying the mecA gene and being PVL-negative.

According to antimicrobial susceptibility testing, all isolates were non-wildtype with respect to the genes for cefoxitin, penicillin, and tetracycline resistance and wild-type with respect to the vancomycin resistance gene (Figure 1 and Table 1).

Notably, three isolates (1.8%) showed resistance to linezolid, with MIC = 8 µg/mL. They contained non-wild-type chloramphenicol, clindamycin, quinupristin/dalfopristin, tetracycline, and tiamulin genes, and two of them also contained non-wild-type erythromycin and trimethoprim genes. Besides, 28 (16.6%) wild-type isolates with respect to linezolid had a MIC = 4 µg/mL.

The frequency of multidrug resistance (MDR) was remarkably high (97.6%), with 55 MDR patterns found (Appendix A). One isolate exhibited an MDR profile (FOX, CLI, ERY, FUS, GEN, KAN, MUP, PEN, SYN, RIF, STR, SMX, TET, TIA, TMP) comprising 11 antimicrobial classes. The most common profile was FOX, CLI, ERY, PEN, SYN, TET, TIA, and TMP, which include seven antimicrobial classes and was found in 27 isolates (16.4%), followed by the profile including FOX, CLI, ERY, PEN, SYN, TET, TIA, found in 17 isolates (10.3%) (Figure 2).

### 2.2. Molecular Characterization of the MRSA Population

Overall, 17 different *spa*-types were identified among the MRSA isolates, i.e., t011, t034, t108, t1451, t15444, t2970, t2971, t4208, t4571, t4885, t567, t6575, t899, and t943, including 3 novel types that were submitted to the Ridom SpaServer platform [18], having been classified as t20098, t20099, and t20100 (Figure 3). The *spa*-type t011 was the most common (52.7%), followed by t108 (27.2%). Furthermore, t011, t034, and t108 belonging to CC398 accounted for 145 isolates (85.8%).

The three isolates resistant to linezolid (MIC = 8 µg/mL) belonged to t011 and showed amplification of the *cfr* gene. However, linezolid-non-wild-type isolates with MIC values equal to the cutoff were negative for the gene.

### 2.3. Whole-Genome Characterization of Linezolid-Resistant MRSA

The genotypic traits of the linezolid-resistant MRSA are summarized in Table 2.

Resistome analysis corroborated the presence of the *cfr* gene and revealed several additional genes in the genome, consistent with the antimicrobial resistance profiles of the isolates. These included β-lactam, encoding the genes *bla*Z and *mec*A, the *fex*A gene, encoding resistance to florfenicol, and genes conferring resistance to tetracycline *tet*(M) and *tet*(K). In addition, one isolate (INIAV_MRSA001) also carried the gene *aad*D conferring resistance to aminoglycosides, and two isolates non-wildtype for the gene for trimethoprim resistance (INIAV_MRSA002 and INIAV_MRSA003) harbored the *dfr*(G) and *dfr*(K) genes, respectively (Table 2). Moreover, all isolates carried efflux pump genes responsible for biocide resistance, namely, *nor*A, *lmr*S, *mep*A, and *sep*A. No chromosomal mutations responsible for antimicrobial resistance were found.

All isolates belonged to the ST398 type, t011 *spa*-type, and SCC*mec*_type_Vc. VirulenceFinder predicted the presence of four virulence factors (*aur*; *hlg*A; *hlg*B; *hlg*C) in all. According to PlasmidFinder, the plasmid replicon sequences rep7a and repUS43 were identified in all isolates. Additionally, several other replicons were found, namely, rep21 and rep22 in INIAV_MRSA001; rep7b and repUS5 in INIAV_MRSA002; and rep22, rep7b, repUS5, and rep16 in INIAV_MRSA003. The plasmids identified by having rep22 carried *tet*(L), *aad*, and/or *dfr*(K), rep7a carried *tet*(K), repUS43 carried *tet*(M), rep7b carried *vga*(A)LC, and in INIAV_MRSA003, rep16 carried the *erm*(B) gene, and repUS5 carried the *fex*(A) gene.

MobileElementFinder did not predict that the *fex*A gene was carried by MGE in INIAV_MRSA001 and INIAV_MRSA002, while the *cfr* gene was predicted to be carried by plasmids. The analysis of the genetic vicinity of the *cfr* gene revealed that all isolates contained the insertion sequence *IS*Sau9 and the transposon TnpR elements flanking the gene. INIAV_MRSA002 also showed the presence of *IS*Bli29 and Tn552 downstream of the TnpR element (Figure 4). The *Isa(B)* gene encoding resistance to lincosamide was carried in the same contig of the *cfr* gene in the INIAV_MRSA001 and MRSA003 strains.

The phylogenetic analysis with the 3 linezolid-resistant isolates from this study and other 23 MRSA ST398 *spa*-type t011 strains recovered from multiple sources, from Belgium and the United Kingdom, revealed that all strains shared between 1 and 519 SNPs. The strains were arranged in two phylogenetic groups, one composed of the isolates from this study (Figure 5, green label), and the other containing the remaining strains (Figure 5, grey label). Therefore, there was no closed genetic relationship between the isolates from this study and the remaining strains (Figure 5). Most strains from Belgium grouped together despite their source, as well as the main strains isolated from horse samples (Figure 5). The most closely related isolates from this study sharing 39 SNPs were INIAV_MRSA002 and INIAV_MRSA003, obtained from samples collected on different days at the same abattoir from animals from different farms.

## 3. Discussion

The present study revealed a very high prevalence of MRSA (98%) in pool nasal swabs collected from healthy, fattening pigs at slaughterhouses in Portugal. Other studies also reported a high prevalence of MRSA in pigs at slaughterhouses and farms in Portugal, ranging from 60 to 99% [3,19,20]. According to the EFSA, the prevalence of MRSA in pigs varies between countries, depending on the sample type, sampling time, antibiotic usage, and legislation implemented at the national level [15].

Globally, 17 different *spa*-types were identified in the MRSA population, t011 and t108 being the most frequent. The high diversity of *spa*-types observed in our study may be attributed to the sample type and the significant number of holdings sampled (*n* = 170) at several slaughterhouses (*n* = 13). In addition, the high variability can be related to international trade; in fact, countries with high trade levels hold a greater diversity of strains [15].

Previous studies also found t011 and t108 as the most common *spa*-types [3,19,20,21]. However, besides t011 and t108, t034 was also identified in our study in 5.8% of the isolates (*n* = 10), following other reports where t034 was one of the most common *spa*-types found in pigs from animal/herds/holdings/slaughterhouses in other European countries [15]. Additionally, strains belonging to three new *spa*-types were found from three farms collected at two slaughterhouses (t20098, t20099, and t20100). The *spa*-type indicates the tandem repeats present in the region X of the *spa* gene, which is very polymorphic. So far, 20,627 different *spa*-types have been registered in the Ridom SpaServer, obtained by the random combination of 832 repeats (last accessed on 11 August 2022). New *spa*-types can be obtained by random rearrangements or by the emergence of new repeats due to genetic mutations or duplication events [22].

Besides the expected resistance to β-lactams, all isolates were also resistant to tetracycline but susceptible to vancomycin, which is in accordance with previous reports [3,19,20]. Overall, 55 different MDR patterns were found; the most common pattern was shared by 27 isolates, with decreased susceptibility to 7 antimicrobial classes, and, notably, one isolate showed resistance to 11 classes. MDR patterns have been observed in LA-MRSA strains originated from the acquisition of genes encoding resistance to several antibiotic families by horizontal gene transfer from other staphylococci and bacteria of human and animal origin [12]. Macrolides, lincosamides, β-lactams, tetracyclines, and sulphonamides are frequently used in pig production as curative, prophylactic, or metaphylactic treatments [23]. The administration of antibiotics can lead to an increase in antimicrobial-resistant bacteria due to their selective pressure exerted on animals and the environment.

The genotype of the linezolid-resistant isolates was in accordance with the susceptibility phenotype, and the resistance genes found, namely, *cfr*, *tet*(K), *tet*(M), *tet*(L), *vga*(A)LC, *erm*(B), *fex*(A), and *dfr*(K), were mainly predicted in plasmids. The *cfr* genes is flanked by *IS*Sau9 (also called *IS*21-558) and the transposon TnpR. The clindamycin exporter gene *Isa*(B) was also located upstream of *IS*Sau9 in two strains, as already described [24]. The *IS*21-558 element, originally identified in the plasmid pSCFS3 recovered from a *S. aureus* strain of pig origin in Germany, was also found in the plasmid pGMI17-006 from an *S. aureus* strain from Denmark [25]. Therefore, *IS*21-558 may be involved in the mobility and dissemination of the multi-resistant gene *cfr* between different staphylococcal species [24,25]. Regardless of our efforts with the in silico analysis, the short-read sequencing performed in this study should be complemented with long-read sequencing to better understand the location of the resistance genes in plasmids. In addition, further efforts will be made to sequence the plasmid carrying the *cfr* gene in the strains from this study to provide information for the scientific community.

In addition, *cfr*-positive MRSA isolates also co-carried some efflux pump genes responsible for biocide resistance (*nor*A, *lmr*S, *mep*A, and *sep*A). Biocides are widely used in animal production as antiseptics and disinfectants, to maintain good hygiene levels in animal holdings and for workers [20]. The three *cfr*-positive isolates were typed as ST398, *spa*-type t011, and SCC*mec*_type_V and harbored four virulence factors, i.e., *aur*, *hlg*A, *hlg*B, and *hlg*C, encoding aureolysin and gamma-hemolysin, respectively.

Previous studies reported the presence of MRSA from ST398 with the same *spa*-type observed in pigs, in humans in close contact with pig farms [3,11,20]. The emergence of *cfr*-mediated linezolid resistance in MRSA from animal origin has also been reported in horses in Germany [26], pigs in Belgium [27], Korea [28], China [29], and Spain [30], and poultry in China [29]. Moreover, *cfr*-positive MRSA strains have been reported sporadically in humans in several European countries, including Italy [31], Portugal [14], Spain [32], as well in hospital outbreaks in Spain [33] and China [34]. Although *cfr*-carrying *Staphylococcus* has already been reported among *S. pseudintermedius* from a dog in Portugal [35], the detection of *cfr*-carrying MRSA ST398 in healthy fattening pigs at slaughterhouses is alarming because of the high zoonotic transmissibility reported for ST398 [11].

Using the conservative cutoffs of 25 wgSNPs proposed by Coll et al. (2020) [36], above which transmission of MRSA within the previous 6 months can be ruled out, we can exclude possible transmission events between the closest strains (INIAV_002 and INIAV_003) from different farms collected in the same abattoir on different days. This strain pair does not comply with the criteria established for the occurrence of a direct transmission (<25 SNPs) because the number of SNP differences was 39 SNPs. However, MRSA strains can persist over time [37], and so those isolates are probably related to the same abattoir. Still, the strains from this study belonging to different farm regions and collected from different abattoirs may indicate that the *cfr* gene is emerging in our country.

Linezolid and vancomycin are the last resource treatments to fight against highly resistant and complicated *S. aureus* infections in humans [13]. Although all the identified linezolid-resistant MRSA strains harboring the *cfr* gene were susceptible to vancomycin, these finding is relevant, as they constitute a potential public health risk. Interestingly, we recently found linezolid-resistant enterococci, including *optrA*- and *poxtA*-positive *Enterococcus faecium* and *optrA*-positive *Enterococcus faecalis*, in pigs from Portugal [38]. These findings highlight the urgency of monitoring linezolid resistance in selected Gram-positive pathogens from animals in Portugal. The emergence of novel resistance genes poses a major threat to human and animal health due to the possibility of horizontal gene transfer from animals to humans and vice-versa through direct contact or the food chain.

The implementation of surveillance and control strategies in the animal and human sectors under the One Health perspective is crucial to better understand the spread of MRSA ST398 in both reservoirs.

## 4. Materials and Methods

### 4.1. Sampling and Bacterial Isolation

One hundred and seventy-one pooled nasal swab samples were collected from pigs sampled at 13 abattoirs across mainland Portugal between October and December 2019. Each pooled sample was composed of nasal swabs from five animals in the same farm, with a total of 170 farms. The samples were sent under refrigeration (4–8 °C) to the National Institute of Agrarian and Veterinary Research (INIAV, IP) for further analyses.

Isolation and identification of MRSA were performed according to the protocol defined by the EU Reference Laboratory for antimicrobial resistance (EURL-AR) [39]. Briefly, the nasal swabs were placed in a pre-enrichment broth containing 6.5% sodium chloride, followed by incubation and plating on the selective chromogenic medium Brilliance MRSA2 agar (Oxoid, Hampshire, UK). The suspected colonies (one single colony by pooled sample) were confirmed to be MRSA by multiplex PCR (*mec*A, *mec*C, *spa*, and *pvl* genes) [40].

### 4.2. Antimicrobial Susceptibility Testing

The antimicrobial susceptibility profile of all isolates was assessed considering 19 antimicrobial agents (Clindamycin, Tetracycline, Rifampin, Streptomycin, Fusidic acid, Penicillin, Chloramphenicol, Kanamycin, Tiamulin, Quinupristin/dalfopristin, Vancomycin, Gentamicin, Trimethoprim, Erythromycin, Ciprofloxacin, Cefoxitin, Linezolid, Mupirocin, and Sulfamethoxazole). The determination of the minimum inhibitory concentrations (MIC) was obtained by the broth microdilution method, using the commercially available 96-well microplate assay Sensititre Staphylococci plate-EUST (Sensititre^®^, Trek Diagnostic Systems, East Grinstead, West Sussex, UK). The results were interpreted according to EUCAST clinical and epidemiological breakpoints (Table 1) [17]. The susceptibility assays used the *S. aureus* strain ATCC 29213 as a quality control. An isolate is defined as MDR if it shows resistance to three or more classes of antimicrobials. The non-wild-type isolates were considered resistant for the determination of multidrug resistance patterns.

### 4.3. Molecular Characterization of the Isolates

Molecular characterization of the isolates based on *spa*-typing was performed. DNA was extracted by the boiling method and tested according to the protocol used by the EURL-AR [41]. The *spa*-type was determined using Ridom SeqSphere+ software v8 (Ridom GmbH, Münster, Germany).

Isolates with MIC ≥ 4 µg/mL for linezolid were tested for the presence of the *cfr* gene by standard PCR, using the primers described by Kehrenberg and Schwarz (2006) [26] for the amplification of a 746 bp fragment. The reactions were carried out in a total volume of 25 µL containing 1 × Gel Load Reaction buffer (NzyTech, Lisbon, Portugal), 2 mM MgCl2 (NzyTech), 400 mM dNTPs (NzyTech), 0.4 μM of each primer, 1U of NZYTaq II DNA polymerase (NzyTech), and 2 μL of DNA. Amplification was performed in a Biometra TOne Thermal Cycler (Analytik Jena, Jena, Germany) with an initial denaturation step at 94 °C for 5 min, followed by 30 cycles at 94 °C for 30 s, 54 °C for 90 s, and 72 °C for 60 s, with a final extension at 72 °C for 10 min. The PCR products were visualized, and images were collected using the UVP BioDoc-It^®^ Imaging System (UVP, Cambridge, UK).

### 4.4. Whole-Genome Sequencing of Linezolid-Resistant Isolates

The genomic DNA of isolates showing resistance to linezolid (MIC > 4 µg/mL) was extracted using the PureLink^®^ Genomic DNA kit, following the Gram-positive bacterial cell lysate protocol (Invitrogen, Carlsbad, CA, USA) and according to the manufacturer’s instructions. DNA was eluted with 50 µL of Tris-HCl buffer, pH 8.5. The quality and quantity of DNA were assessed using a spectrophotometer (Nanodrop^®^ 2000, Thermo Scientific, Waltham, MA, USA) and a QuantusTM Fluorometer with the QuantiFluor^®^ ONE dsDNA Dye kit (Promega, Madison, WI, USA), according to the manufacturer’s recommendations. Library preparation and DNA sequencing were performed by Novogene Europe, UK, using the Illumina HiSeq sequencing technology (NovaSeq 6000 S2 PE150 XP sequencing mode). The nucleotide sequences were deposited in, the European Nucleotide Archive (ENA) [42] with the accession numbers ERS6142034, ERS6142035, and ERS6142036.

Raw data quality was assessed by FastQC [43] and low-quality sequencing data and adapter sequences were removed using Trimmomatic v0.27 [44] with default settings. All pre-processed reads were assembled with SPAdes v3.12.0 [45], and the assembly stats were calculated using QUAST-5.0.2 [46]. Bioinformatics analyses were performed after removing the contigs with sizes lower than 500 bp.

Acquired antimicrobial resistance genes and chromosomal point mutations, plasmid replicons, multi-locus sequence type (MLST), identification of virulence genes, pathogenicity, *spa*-type, and staphylococcal cassette chromosome mec (SCCmec) elements were assessed using ResFinder v4.0 (command line, 90% threshold for %ID/60% minimum length) [47,48], PlasmidFinder (command line, 95% threshold for %ID) [49], MLST (command line) [50], VirulenceFinder (command line, 90% threshold for %ID/60% minimum length) [51], PathogenFinder v1.1 [52], SpaTyper v1.0 [53], and SCCmecFinder v1.2 (90% threshold for %ID/60% minimum length) [54,55,56], respectively. MobileElementFinder v1.0.3 was used to identify mobile genetic elements and their relation to antimicrobial resistance genes and virulence factors [57]. The Comprehensive Antibiotic Resistance Database (CARD) [58] was used to complement the characterization of the isolates genomic content.

For identifying the genetic platform of the *cfr* gene, the contigs containing this gene were annotated using Prokka v1.14.6 [59], followed by analysis with Artemis [60], EasyFig v2.2.5 [61], and Basic Local Alignment Search Tool (BLAST) from NCBI website.

A phylogenetic analysis based on single-nucleotide polymorphisms (SNPs) present in the genomes, using CSI Phylogeny v.1.4 (10 reads of minimal depth at SNP positions, 10% minimal relative depth at SNP positions of, 10 bp of minimal distance between SNP, minimal SNP quality of 30, minimal read mapping quality of 25, and a minimal Z-score of 1.96) [62] from the CGE website was conducted with the three MRSA from this study and other 23 MRSA ST398 strains from multiple sources from Belgium and the United Kingdom [63]. *Staphylococcus aureus* strain ISU926 isolate ST398 (accession number CP017091.1) was used as the reference genome. The graphical representation and tree annotation were performed using iTOL, Interactive Tree Of Life v6. [64].

## Figures and Tables

**Figure 1 antibiotics-11-01439-f001:**
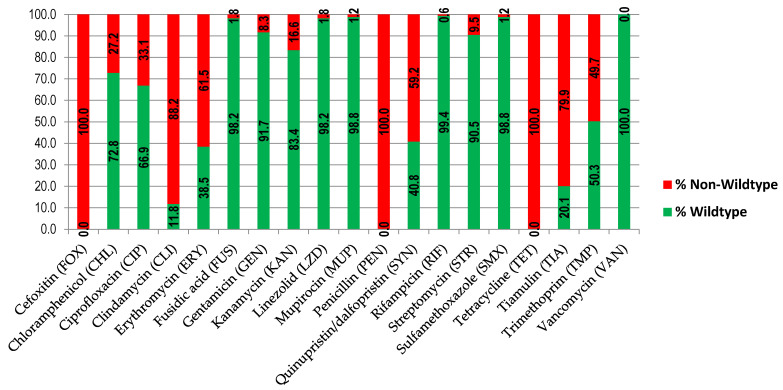
Antibiotic non-susceptibility resistance distribution of the MRSA isolates (*n* = 169). Each bar represents the phenotypic result obtained for each antibiotic according to the ECOFF MIC values. The red bars represents the percentages of the non-wildtype isolates, and the green bars represents the percentages of the wild-type isolates for the respective antibiotic.

**Figure 2 antibiotics-11-01439-f002:**
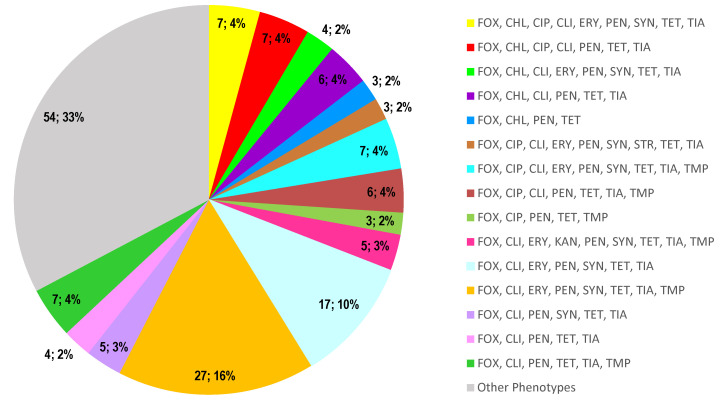
Distribution of the MDR profiles, showing the 15 most common phenotypes of the MDR strains (*n* = 165). The number and the percentage of isolates are indicated in the graph. FOX-Cefoxitin, CHL-Chloramphenicol, CIP-Ciprofloxacin, CLI-Clindamycin, ERY-Erythromycin, FUS-Fusidic acid, GEN-Gentamicin, KAN-Kanamycin, LZD-Linezolid, MUP-Mupirocin, PEN-Penicillin, SYN-Quinupristin/dalfopristin, RIF-Rifampicin, STR-Streptomycin, SMX-Sulfamethoxazole, TET-Tetracycline, TIA-Tiamulin, TMP-Trimethoprim, VAN-Vancomycin.

**Figure 3 antibiotics-11-01439-f003:**
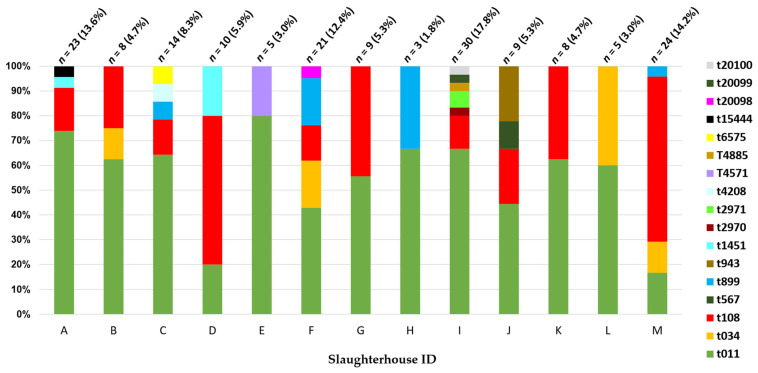
Distribution of *spa*-types among the MRSA groups isolated from samples collected from the 13 slaughterhouses (*n* = 169). Each bar represents the *spa*-types obtained from a slaughterhouse. The number and the percentage of isolates are shown for each bar.

**Figure 4 antibiotics-11-01439-f004:**
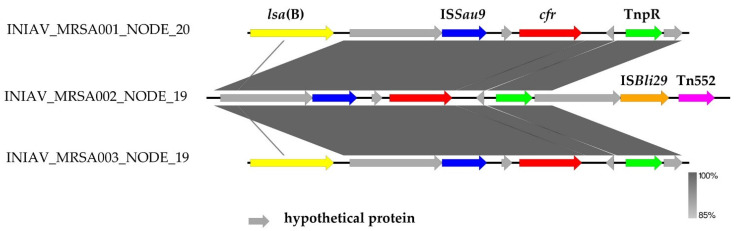
Genetic platforms of the *cfr* genes in the MRSA isolates determined using EasyFig. This figure represents the genomic environment of the *cfr* genes, regarding mobilization elements (insertion sequences and transposons) and other antibiotic determinants. The gray area represents the blast identities, and the percentage of identity is indicated in the legend.

**Figure 5 antibiotics-11-01439-f005:**
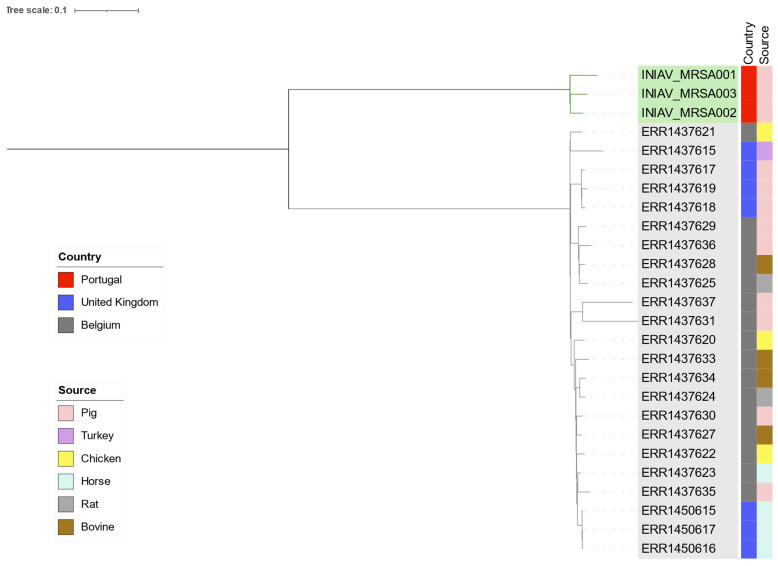
Phylogenetic analysis based on core genome single-nucleotide polymorphisms (SNPs) using CSI Phylogeny v.1.4 of 26 MRSA ST398, t011 strains from multiple sources and different countries. Graphic representation using the iTol interactive tree of life, showing the geographic location, source, and genetic distance of the isolates. In green are the isolates from this study. ERR*******, accession numbers of the strains used in this analysis.

**Table 1 antibiotics-11-01439-t001:** Clinical and epidemiological breakpoints, MIC50, MIC90, and frequency of resistance of the MRSA isolates.

Antimicrobial	Clinical MIC Breakpoints [17] (mg/L)	Epidemiological MIC Breakpoints [17] (mg/L) (T)	MIC50 (mg/L) (*n* = 169)	MIC90 (mg/L) (*n* = 169)	% DS (N)
Cefoxitin (FOX)	4	NA	>16	>16	100.0 (169)
Chloramphenicol (CHL)	8	NA	16	64	27.2 (46)
Ciprofloxacin (CIP)	1	1	0.5	>8	33.1 (56)
Clindamycin (CLI)	0.25	0.25	>4	>4	88.2 (149)
Erythromycin (ERY)	2	1	>8	>8	61.5 (104)
Fusidic acid (FUS)	1	0.5	≤0.5	≤0.5	1.8 (3)
Gentamicin (GEN)	2	(2)	≤1	≤1	8.3 (14)
Kanamycin (KAN)	8	NA	≤4	32	16.6 (28)
Linezolid (LZD)	4	4	2	4	1.8 (3)
Mupirocin (MUP)	NA	(1)	≤0.5	≤0.5	1.2 (2)
Penicillin (PEN)	0.125	NA	>2	>2	100.0 (169)
Quinupristin/dalfopristin (SYN)	2	NA	2	4	59.2 (100)
Rifampicin (RIF)	0.06	(0.03)	≤0.016	≤0.016	0.6 (1)
Streptomycin (STR)	NA	NA	8	16	9.5 (16)
Sulfamethoxazole (SMX)	NA	NA	≤64	≤64	1.2 (2)
Tetracycline (TET)	2	1	>16	>16	100.0 (169)
Tiamulin (TIA)	NA	NA	>4	>4	79.9 (135)
Trimethoprim (TMP)	NA	(2)	≤2	>32	49.7 (84)
Vancomycin (VAN)	2	2	≤1	≤1	0.0 (0)

DS, Decreased susceptibility; NA, Not Applicable; (T), Tentative ECOFF; N, number of DS isolates.

**Table 2 antibiotics-11-01439-t002:** Whole-genome characterization of linezolid-resistant MRSA (*n* = 3).

Feature	INIAV_MRSA001	INIAV_MRSA002	INIAV_MRSA003
Sampling date	October 22	November 7	November 20
Farm region	AML (Peninsula de Setubal)	Alentejo	Alentejo Litoral
Slaughterhouse ID (region)	A (AML, Península de Setúbal)	K (North region, Ave)	K (North region, Ave)
Phenotype	FOX, CHL, CLI, ERY, LZD, PEN, SYN, TET, TIA, TMP	FOX, CHL, CLI, ERY, LZD, PEN, SYN, TET, TIA, TMP	FOX, CHL, CLI, LZD, PEN, SYN, TET, TIA, TMP
Antibiotic resistance genes	*aadD, blaZ, mecA, lsa(B), **cfr**, fexA, tet(M), tet(L), tet(K)*	*blaZ, mecA, vga(A)LC, **cfr**, fexA, tet(M), tet(K), dfrG*	*blaZ. mecA, erm(B), Isa(B), **cfr**, vga(A)LC, fexA, tet(M), tet(L), tet(K), dfrK*
Biocide resistance genes	*norA, lmrS, mepA, sepA*	*norA, lmrS, mepA, sepA*	*norA, lmrS, mepA, sepA*
SCCmec elements	SCC*mec*_type_Vc(5C2&5)	SCC*mec*_type_Vc(5C2&5)	SCC*mec*_type_Vc(5C2&5)
Virulence genes	*aur; hlgA; hlgB; hlgC*	*aur; hlgA; hlgB; hlgC*	*aur; hlgA; hlgB; hlgC*
Plasmid replicons	rep21, rep22, rep7a, repUS43	rep7a, rep7b, repUS43, repUS5	rep16, repUS5, rep22, rep7a, rep7b, repUS43
Pathogenicity (%)	97.9	97.7	97.7
*Spa*-type	t011	t011	t011
MLST	ST398	ST398	ST398

FOX-Cefoxitin, CHL-Chloramphenicol, CIP-Ciprofloxacin, CLI-Clindamycin, ERY-Erythromycin, FUS-Fusidic acid, GEN-Gentamicin, KAN-Kanamycin, LZD-Linezolid, MUP-Mupirocin, PEN-Penicillin, SYN-Quinupristin/dalfopristin, RIF-Rifampicin, STR-Streptomycin, SMX-Sulfamethoxazole, TET-Tetracycline, TIA-Tiamulin, TMP-Trimethoprim, VAN-Vancomycin, AML-Área Metropolitana de Lisboa.

## Data Availability

The data that support the findings of this study are available within the article. The whole-genome sequence data were deposited in the European Nucleotide Archive with the accession numbers ERS6142034, ERS6142035, and ERS6142036.

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
