# Peer review of "Emergence of Cfr-Mediated Linezolid Resistance among Livestock-Associated Methicillin-Resistant Staphylococcus aureus (LA-MRSA) from Healthy Pigs in Portugal"

_antibiotics, 2022, doi:10.3390/antibiotics11101439_

Round 1

Reviewer 1 Report

The manuscript under review provides a detailed characterization of 171 isolates of livestock-associated methicillin-resistant Staphylococcus aureus (LA-MRSA) originating from nasal swabs of pigs in farms in Portugal. For all isolates, phenotypic sensitivity/resistance to selected antimicrobials was determined by the microdilution method, resistance profiles were identified, and molecular analysis was performed using spa-typing. In addition, whole-genome analysis was performed for three linezolid-resistant MRSA isolates. The chosen methods of characterizing the isolates areappropriate, the results are clearly presented and discussed.

I propose to publish the manuscript in the journal Antibiotics in its original form without changes.

Author Response

Dear reviewer

We thank you for the revision work and kind comments.

Best regards

Reviewer 2 Report

The manuscript is scientifically interesting and with relevant implications especially for public health.

LA-MRSA CC398 represents a zoonotic risk. Especially for people working in close contact with livestock: farmers, veterinarians, abattoir workers, and people living in areas with a high livestock density.

Furthermore, in the work it appears clear that the global trend of increasing antimicrobial resistance in S. aureus is becoming problematic and this critical point is correctly emphasized.

The sentence that I report directly from the publication is particularly significant and important: "the emergence of novel resistance genes poses a major threat to human and animal health due to horizontal gene transfer from animals to humans and vice-versa through direct contact or the food chain. The implementation of surveillance and control strategies in the animal and human sectors under a One Health perspective is crucial to better understand the spread of MRSA ST398 in both reservoirs. "

I completely agree with the sentence and with the global aim of the study.

Just one little suggestion, I would modify the phrase:

237 The administration of antibiotics can lead to the increase of antimicrobial-resistant bacteria, due to its selective pressure on the (gut) animals and environment. 238

with the phrase:

237 Administration of antibiotics can lead to an increase in antimicrobial resistant bacteria, due to its selective pressure exerted on animals and the environment. 238

Congratulation and Best Regards

Author Response

Dear reviewer

We thank you for the revision work and kind comments.

Your suggestion improves our study.

Best regards

Reviewer 3 Report

1. The introduction section should be shortened and simplified; some contents can be moved to discussion, such as lines 59-66.

2. Figure 1: The number of non-wildtype and wildtype should also be proved in each column to make it easier to follow.

The same thing for Tale 1: the % DS may describe as “positive number/total number, %”.

Please also do this for Figure 2 and Figure 3, including both the positive number and percentage.

3. Figure 2: What figure 2 do the authors want to say? How to read the data? I was confused with the graphic and “present in three or more isolates.

Please include the description of the figures in the figure legend, not just the title, for all the figures (Figure 1 to Figure 5) in the manuscript.

4. Table 2 and Figure 3: The characterization of these three isolates’ whole genome is impressive; how about comparing them to the other control isolates, like the ATCC strain or strains reported in the other area or countries? Then, you may find some gene shifting or functional changes. It may not necessarily include so many strains as it was performed in Figure 5.

5. Figure 5: Was the phylogenetic tree generated with the whole genome or just cfr gene? If not the whole genome, then why not use the whole genome for comparison?

The numbers next to each node of the phylogenetic tree, representing the measure of support for the node, should be included in the figure. 

The details of what technique was used should also be in the figure legend.

6. Is the whole genome sequencing data published or will publish is somewhere else? I was just surprised the authors didn’t perform a deep analysis of this data.

Author Response

Dear reviewer

We thank the reviewer for the revision work and suggestions to improve our study.

Please find below our reply to your Comments and Suggestions.

Best regards,

  1. The introduction section should be shortened and simplified; some contents can be moved to discussion, such as lines 59-66.

Reply: We acknowledge the point of view of the reviewer. Nevertheless, lines 59-66 have valuable information for our work's context. Moreover, the introduction section is 43 lines long, which is not so extensive. Based on these arguments, we are maintaining these sentences for the reader’s best interest.

  1. Figure 1: The number of non-wildtype and wildtype should also be proved in each column to make it easier to follow.

The same thing for Tale 1: the % DS may describe as “positive number/total number, %”.

Please also do this for Figure 2 and Figure 3, including both the positive number and percentage.

Reply: The information related to the percentage of the non-wildtype population from figure 1 corresponds to the %DS (resistant isolates), which is shown in table 1. Therefore, the percentage of non-wildtype and wildtype was added to figure 1, and the number of isolates was added to table 1, as suggested by the reviewer. Figure 3 was also updated, showing the percentages of strains by the slaughterhouse.

  1. Figure 2: What figure 2 do the authors want to say? How to read the data? I was confused with the graphic and “present in three or more isolates.

Please include the description of the figures in the figure legend, not just the title, for all the figures (Figure 1 to Figure 5) in the manuscript.

Reply: We acknowledge and agree with the reviewer's suggestion, as adding the description of the figures increases the interpretation of the information and makes it easier to understand the figures. We also redo figure 2 to simplify the data reading.

  1. Table 2 and Figure 3: The characterization of these three isolates’ whole genome is impressive; how about comparing them to the other control isolates, like the ATCC strain or strains reported in the other area or countries? Then, you may find some gene shifting or functional changes. It may not necessarily include so many strains as it was performed in Figure 5.

Reply: Staphylococcus aureus strain ISU926 isolate ST398 (accession number CP017091.1) was used as the reference genome for the phylogenetic analysis because it is an LA-MRSA ST398 strain isolated from a pig, and SNPs were determined according to the reference genome.

We haven’t performed a deeper genomic analysis as suggested by the reviewer (gene shifting or functional changes). In this study, we were only searching for the resistance genes and their relationship with mobile genetic elements responsible for the dissemination of genes. However, we expect to perform nanopore sequencing to complement the WGS analysis for a deeper analysis of these strains. Moreover, we also wish to sequence some linezolid-susceptible strains from some slaughterhouses for comparison. Nonetheless, these works are delayed, and we decided that these data were important to share with the scientific community as fast as we could. 

  1. Figure 5: Was the phylogenetic tree generated with the whole genome or just cfr gene? If not the whole genome, then why not use the whole genome for comparison?

The numbers next to each node of the phylogenetic tree, representing the measure of support for the node, should be included in the figure.

The details of what technique was used should also be in the figure legend.

Reply: The phylogenetic tree was generated with the SNPs from the whole genome as stated in the material in methods section (lines 372-379). The figure and the legend were rephrased to include additional information.

  1. Is the whole genome sequencing data published or will publish is somewhere else? I was just surprised the authors didn’t perform a deep analysis of this data.

Reply: Whole genome sequence data are deposited at European Nucleotide Archive with the accession numbers ERS6142034, ERS6142035, and ERS6142036 as stated in material and methods section, we also added this information at section Data Availability Statement.

As replied to question 4, we expect to soon perform nanopore sequencing to complement the WGS analysis and perform a deeper analysis of these strains and respective plasmids.

Round 2

Reviewer 3 Report

The authors addressed most of my concerns.